# Changes in Expression of Specific mRNA Transcripts after Single- or Re-Irradiation in Mouse Testes

**DOI:** 10.3390/genes13010151

**Published:** 2022-01-15

**Authors:** Kenta Nagahori, Ning Qu, Miyuki Kuramasu, Yuki Ogawa, Daisuke Kiyoshima, Kaori Suyama, Shogo Hayashi, Kou Sakabe, Takayuki Yoshimoto, Masahiro Itoh

**Affiliations:** 1Department of Anatomy, Tokyo Medical University, Tokyo 160-8402, Japan; kenta-n@tokyo-med.ac.jp (K.N.); kitaoka@tokyo-med.ac.jp (M.K.); yogawa@tokyo-med.ac.jp (Y.O.); itomasa@tokyo-med.ac.jp (M.I.); 2Department of Anatomy, Division of Basic Medical Science, Tokai University School of Medicine, Kanagawa 259-1193, Japan; kiyoshima@tokai.ac.jp (D.K.); suyama@is.icc.u-tokai.ac.jp (K.S.); sho5-884@umin.ac.jp (S.H.); sakabek@tokai-u.jp (K.S.); 3Immuno Regulation, Institute of Medical Science, Tokyo Medical University, Tokyo 160-8402, Japan; yoshimot@tokyo-med.ac.jp

**Keywords:** irradiation, germ cell differentiation, Sertoli tight junctions, anti-sperm antibody

## Abstract

Alkylating agents and irradiation induce testicular damage, which results in prolonged azoospermia. Even very low doses of radiation can significantly impair testis function. However, re-irradiation is an effective strategy for locally targeted treatments and the pain response and has seen important advances in the field of radiation oncology. At present, little is known about the relationship between the harmful effects and accumulated dose of irradiation derived from continuous low-dose radiation exposure. In this study, we examined the levels of mRNA transcripts encoding markers of 13 markers of germ cell differentiation and 28 Sertoli cell-specific products in single- and re-irradiated mice. Our results demonstrated that re-irradiation induced significantly decreased testicular weights with a significant decrease in germ cell differentiation mRNA species (*Spo11*, *Tnp1*, *Gfra1, Oct4, Sycp3, Ddx4*, *Boll, Crem, Prm1,* and *Acrosin*). In the 13 Sertoli cell-specific mRNA species decreased upon irradiation, six mRNA species (*Claudin-11,*
*Espn, Fshr, GATA1, Inhbb**,* and *Wt1*) showed significant differences between single- and re-irradiation. At the same time, different decreases in Sertoli cell-specific mRNA species were found in single-irradiation (*Aqp8*, *Clu*, *Cst12*, and *Wnt5a*) and re-irradiation (*Tjp1, occludin,*
*ZO*-1, and *ZO*-2) mice. These results indicate that long-term aspermatogenesis may differ after single- and re-irradiated treatment.

## 1. Introduction

Radiotherapy and chemotherapy are commonly used as treatments for cancer. However, their use is associated with serious side effects, including temporary or permanent infertility. Recently, we demonstrated that traditional Japanese medicine, Goshajinkigan, can rescue mice aspermatogenesis after chemotherapy and radiotherapy [1,2]. In these two studies, we found that the busulfan- and irradiation-induced impaired reproductive functions were related to the different immune-pathophysiological conditions. In particular, Goshajinkigan was found to have different therapeutic effectiveness in response to this aspermatogenesis. Although supplementation with Goshajinkigan in irradiation-mice was found to rescue the disrupted inter-Sertoli tight junctions via the normalization of *claudin11*, *occludin*, and *ZO-1* expression, as well as reduce serum anti-germ cell autoantibodies, the recovery of spermatogenesis (day 150) was more pronounced than that of Goshajinkigan supplementation in busulfan experiments (day 120) [1,2]. Sertoli cell tight junctions are a physical barrier in the blood-testis barrier (BTB) [3,4,5], resulting in the provision of an environment within the adluminal region that is both immune protective and of a controlled biochemical nature, which allows meiotic and post-meiotic germ cell maturation to proceed. The breakdown of BTB by irradiation can cause the germ cell autoantigens inside BTB to leak out repeatedly. The subsequent autoimmune reaction against germ cells after irradiation may affect the delayed recovery of spermatogenesis damage [6].

Total body irradiation, used as conditioning for bone marrow transplants, is associated with appreciable gonadal toxicity. Previous studies have shown that 99.5% of men who received 12 Gy total body irradiation experience permanent infertility [7], with irradiation doses as low as 5–6 Gy causing a decrease in spermatogenesis in the seminiferous tubules of mice [8,9]. With significant technological advances in the field of radiation oncology, integrated body imaging with accurate treatment delivery methods, such as stereotactic body radiotherapy, could improve the efficacy, shorten the overall treatment time, and potentially reduce treatment-related toxicities, indicating that re-irradiation is an effective treatment for local control and pain response [10]. However, little is known about the relationship between the harmful effects and accumulated dose derived from continuous low-dose radiation exposure. In the present study, we examined the impaired testicular functions induced by single- or re-irradiation in mice to address the immune-pathophysiological differences between these two treatment types.

## 2. Materials and Methods

### 2.1. Animals

C57BL/6J male mice at 4 weeks of age (weighing 16–20 g) were purchased from SLC (Shizuoka, Japan) and kept in the Laboratory Animal Center of Tokyo Medical University (Tokyo Medical University Animal Committee; no. H28016-01042017). Mice were maintained at a temperature of 22–24 °C and relative humidity of 50–60% with a 12-h light–dark cycle. 

### 2.2. Experimental Design

In the present study, we examined the impaired testicular functions induced by low-dose radiation exposure in mice to address the immune-pathophysiological differences between single- or re-irradiation. Mice were irradiated with 4 or 6 Gy using a 60Co gamma ray unit (MBR-1520A-TWZ; HITACHI KE Systems Ltd, Tokyo, Japan). Radiation was administered to the body without anesthesia. Then, the study mice were randomly divided into the following two experimental groups: (1) group I: mice in the single-irradiated group (*n* = 20) were administered a single dose of 4 Gy (4 GyI; *n* = 10), or 6 Gy (6 GyI; *n* = 10); (2) group II: mice in the re-irradiated group (*n* = 20) were exposed to X-ray radiation with 1 Gy four times (4 GyII; *n* = 10) or six times (6 GyII; *n* = 10) with an interval of one day. The doses were monitored during each irradiation procedure. The general condition, food intake, and body weight were documented for all mice one week after the final irradiation. The protocol used in the present study was approved by the Institutional Animal Care and Use Committee at the Dongnam Institute of Radiological and Medical Sciences and the Tokyo Medical University Animal Committee (no. H28016). The animals were maintained in accordance with the Guidelines of the National Institutes of Health for Animal Experiments. Non-irradiated mice (ConI and ConII; *n* = 20) were housed on shelves in the same facility and shielded from radiation.

To examine spermatogenesis after irradiation, the mice from each group (*n* = 10) were anesthetized with pentobarbital (65 mg/kg body weight), after which the testes and epididymides were removed for gravimetry one week after the final irradiation exposure.

### 2.3. Histological Examination of the Testes

The testes of mice from each group (*n* = 5) were fixed with Bouin’s solution and embedded in plastic (Technovit7100; Kulzer & Co., Wehrheim, Germany). Sections (5 μm) were obtained at 15–20-μm intervals and were stained with Gill’s hematoxylin and 2% eosin Y (Muto PC, Tokyo, Japan) for observation under a light microscope.

### 2.4. Detection of Serum Anti-Germ Cell Antibodies

To prepare the antigens, testes from normal mice (8-week-old; *n* = 5) were homogenized in carbonate–bicarbonate buffer (Sigma-Aldrich, MO, USA) and germ cell concentrations were adjusted to 50 µg/200 µL. The antigens were added to 96-well microtiter plates (Nunc 96-microwell plate; Thermo Fisher Scientific, Kanagawa, Japan) and incubated at 37 °C for 30 min. After removing the coating solution, the wells were washed in triplicate with PBS. Then, 200 µL of sera (1:160 dilution) from each group of mice (*n* = 10) was added to micro-ELISA wells in duplicate and incubated for 2 h at room temperature. Next, the wells were washed three times with PBS–Tween-20 (0.05% *v/v*) and incubated for 2 h at room temperature with 200 µL of HRP-conjugated goat anti-mouse IgM (1:10,000) (Cappel). One o-phenylenediamine dihydrochloride (OPD) tablet dissolved in 20 ml of water with one phosphate–citrate tablet (Sigma-Aldrich) was used as a soluble substrate to detect peroxidase activity. The wells were washed five times with PBS–Tween-20, and 200 µL of freshly prepared OPD solution was added to each well. After 30 minutes, the sample absorbance at 450 nm was measured using MPR-A4 (Tokyo, Japan) [11].

For the detection of serum anti-germ cell antibodies, normal testes from normal 8-week-old mice were placed in OCT compound (Miles Laboratories, IL, USA), frozen in liquid nitrogen, and stored at −80 °C until further use. Sections (6 µm) were cut using a cryostat (CM1900; Leica, Wetzlar, Germany), dried in the air, fixed in 95% ethanol for 10 min at −20 °C, rinsed in PBS, and then incubated with 50-fold serial dilutions of the collected serum samples from experimental mice of each group (*n* = 10) for 60 min at RT. After rinsing in PBS, the cryostat sections were incubated for 60 min with HRP-conjugated goat anti-mouse IgM (1:500 dilution) (ZyMax, CA, USA) at room temperature. After washing with PBS, the HRP-binding sites were detected with 0.05% DAB and 0.01% H_2_O_2_.

### 2.5. Analysis of the Specific mRNA Species in Testes Using Real-Time RT-PCR 

Testes from mice in each group (*n* = 5) were examined. The iCycler thermal cycler (Bio-Rad, Hercules, CA, USA) was used to perform the PCR reactions, and the mixtures were stored at −80 °C until analysis. Real-time RT-PCR was performed on 3 ng cDNA using the validated SYBR Green gene expression assay in combination with SYBR Premix Ex Taq^TM^ (TaKaRa, Bio Inc., Ohtsu, Japan) for measuring 13 germ cell genes (*Stra8, Spo11, Tnp1,*
*cKit, Gfra1, Oct4, PLZF, Sycp3, Ddx4, Boll, Crem, Prm1*, and *Acrosin*), 28 Sertoli cell genes (*Amh, Aqp8, Ccnd2, Clu,*
*Claudin-11,*
*Cst12, Cst9, Dhh, Espn, Fshr, Fyn, GATA1,*
*Inhba**,*
*Inhbb**, Msi1,*
*Occlidin,*
*Rhox5, Testin, Spata2,*
*shbg**,Sox9, sympk, Tjp1, Trf, Wt1,*
*Wnt5a, ZO-1*, and *ZO-2*) and GAPDH. Quantitative real-time PCR was performed in duplicate using the Thermal Cycler Dice Real-time System TP800 (TaKaRa). The Thermal Cycler Dice Real-time System software Ver.5.1.1 (TaKaRa) was used to analyze the data, and the comparative C_t_ method (2^-∆∆Ct^) was used to quantify gene expression levels. The results are expressed relative to the amount of GAPDH transcript used as an internal control. Relative mRNA intensity was calculated, and the expression in the control group for each point was normalized to 1. Data are presented as the mean ± standard deviation (SD). The primers used in the analysis are listed in Table 1.

### 2.6. Statistical Analysis

ANOVA and Tukey-Kramer post-hoc tests were used to analyze the differences between multiple groups. Statistical significance was set at *p* < 0.05.

## 3. Results

### 3.1. Histopathological Changes after Irradiation Exposure

To examine histopathological changes after irradiation, we documented the testes’ weight and observed the testes sections of mice from each group under a light microscope. Non-irradiated mice showed spermatogenesis maturity (Figure 1A) and increased body and testis weights (Figure 1B) from the ages of five weeks (ConI) to six weeks (ConII). On the other hand, atrophic seminiferous tubules with disrupted spermatogenesis were observed in the testes of all irradiated mice, with significantly decreased testicular weights in the mice of group II after re-irradiation with 4 or 6 Gy. In contrast, no significant differences in the body weight were observed between group I by single-irradiated (4 and 6 GyI), group II by re-irradiation (4 and 6 GyII), and the non-irradiated groups (ConI and ConII) (Figure 1B). These data showed re-irradiation significantly decreased testicular weight with atrophic seminiferous tubules, as well as disrupting spermatogenesis in the testes.

### 3.2. Effect of Irradiation Exposure on Levels of mRNA Transcripts Encoding Markers of Germ Cell Differentiation Products

For the detection of the effect of single- and re-irradiation on testicular spermatogenesis, we compared the expression levels of mRNA species encoding the spermatogonial marker Stra8, the spermatocyte marker of Spo11, the spermatid marker Tnp1 (Figure 2A), premeiotic cells (cKit, Gfra1, Oct4, PLZF, Sycp3, and Ddx4), and meiotic and postmeiotic cells (Boll, Crem, Prm1, and Acrosin) (Figure 2B) by real-time RT-PCR analysis in each group [12,13,14,15,16,17,18,19,20]. The levels of almost all examined spermatogenesis markers, excluding cKit, were significantly reduced in group II, compared to five spermatogenesis markers (Stra8, Spo11, Oct4, Sycp3, and Boll) that were significantly reduced in group I (Figure 2A,B). Furthermore, the spermatocyte marker of Spo11, the spermatid marker Tnp1, and eight differentiation markers (Gfra1, Oct4, Sycp3, Ddx4, Boll, Crem, Prm1, and Acrosin) were significantly reduced in group II compared to those in group I. These data showed the spermatocyte marker, the spermatid marker, and some mRNA species of germ cell differentiation markers were significantly affected by re-irradiation.

### 3.3. Effect of Irradiation on Levels of mRNA Transcripts Encoding Markers of Sertoli Cell-Specific Products

To examine the Sertoli cell function regulated by irradiation-induced germ cell depletion, the levels of 28 Sertoli cell-specific mRNA species were measured in each group (Figure 3). Thirteen mRNA species (*Amh, Claudin-11, Dhh, Espn, Fshr, Fyn, GATA1, Inhbb, Msi1, Spata2, Sox9, Trf,* and *Wt1*) showed significantly decreased expression (Figure 3A) in response to irradiation (group I and II) and five mRNA species (*Ccnd2, Cst9, Rhox5, Shbg,* and *Testin*) showed increased expression in response to irradiation (group I and II) (Figure 2B). Furthermore, four mRNA species (*Aqp8, Clu, Cst12,* and *Wnt5a*) showed a significant decrease in levels only in group I (single irradiation) but not in group II (re-irradiation) (Figure 3C). In contrast, the other four mRNA species (Tjp1, occlidin, ZO-1, and ZO-2) showed a significant decrease in levels only in group II after re-irradiation but not in group I after single irradiation (Figure 3D). No significant changes in the expression levels of the remaining two mRNA species (Inhba and symp) were observed after irradiation treatment (group I and II) (Figure 3E). The above data showed that single- and re-irradiation treatment induced different increased or decreased in Sertoli cell-specific mRNA species, and six mRNA species (*Claudin-11, Espn, Fshr, GATA1, Inhbb,* and *Wt1*) showed significant differences between single- and re-irradiation in all 13 decreased Sertoli cell-specific mRNA species expression after irradiation.

### 3.4. Detection of Serum Anti-Germ Cell Antibodies

For the detection of serum anti-germ cell antibodies in mice from each group, we measured the level of autoantibodies by ELISA and immunohistochemistry analysis. As shown in Figure 4A, the serum IgM levels in mice after irradiation treatment were significantly higher than those in control mice. Additionally, a significant difference was observed in the mice in group 6 GyI and 6 GyII. The serum IgM levels in the mice in group 6 GyII were significantly lower than those in group 6 GyI.

Upon assessing the reaction of sera with normal frozen seminiferous tubule section, anti-germ cell antibodies were detected in all irradiation groups (group I and II), but not in the normal group (Figure 4B). Serum autoantibodies preferentially reacted with mature spermatids and spermatozoa; concurrently, faint immunostaining was detected in spermatogonia and immature spermatids.

## 4. Discussion

In this study, we examined 13 markers of germ cell differentiation and 28 Sertoli cell-specific products in single- or re-irradiated mice to clarify the pathophysiological differences between single- and re-irradiation. We found that re-irradiation significantly decreased testicular weight with atrophic seminiferous tubules, as well as disrupting spermatogenesis in the testes (Figure 1). Furthermore, the spermatocyte marker of *Spo11*, the spermatid marker *Tnp1*, and eight mRNA species of differentiation markers (*Gfra1, Oct4, Sycp3, Ddx4**, Boll, Crem, Prm1,* and *Acrosin*) were significantly reduced in re-irradiated mice compared to single-irradiated mice (Figure 2A,B). At the same time, irradiation treatment induced a decrease in Sertoli cell-specific mRNA species after single irradiation *(**Aqp8, Clu, Cst12*, and *Wnt5a**)* (Figure 3C) and after re-irradiation (*Tjp1, occlidin, ZO*-1, and *ZO*-2) (Figure 3D). In all 13 Sertoli cell-specific mRNA species in which expression was decreased, six mRNA species (*Claudin-11,*
*Espn, Fshr, GATA1, Inhbb**,* and *Wt1*) showed significant differences between single- and re-irradiation. These decreased Sertoli cell-specific mRNA species were accompanied by increased serum anti-germ cell antibody levels. In all five Sertoli cell-specific mRNA species in which expression was increased, only *Testin* showed a significant difference between single- and re-irradiation. 

Infertility and gonadal dysfunction caused by cancer treatments have become a significant concern for cancer survivors, with substantial improvements in anticancer therapy. It is well known that spermatogenesis is a complex and tightly regulated process that leads to the continuous production of male gametes, i.e., spermatozoa. The depletion of spermatogonial stem cells is the apparent cause of infertility in oncological patients after chemo- and radiotherapy [1,21,22,23], and may play a leading role in the oligo- and azoospermia of infertility patients [24,25]. Meiotic and post-meiotic germ cells (spermatocytes and especially spermatids) are much less sensitive to cytotoxic treatment and radiation than spermatogonia [26]. Although radiotherapy-induced aspermatogenesis has been extensively studied, no study has yet addressed the differences between single- and re-irradiation-induced male infertility. In this study, we found that some atrophic seminiferous tubules with disrupted spermatogenesis were observed in the testes of all irradiated mice (Figure 1). In particular, we noted that the testicular weights, the spermatocyte marker of *Spo11*, the spermatid marker *Tnp1*, the pre-meiotic markers (*Gfra1, Oct4, Sycp3,* and *Ddx4*), and all the meiotic and post-meiotic markers (*Boll, Crem, Prm1,* and *Acrosin*) were significantly reduced in re-irradiated mice compared to single-irradiated mice (Figure 1 and Figure 2). These results demonstrate that re-irradiation damaged not only spermatogonia but also meiotic and post-meiotic germ cells (spermatocytes and spermatids). In particular, all the meiotic and post-meiotic markers examined in this study were significantly reduced by re-irradiation, indicating that the long-term aspermatogenesis in patients with longer survival may differ between single- and re-irradiated treatments.

In this study, we further examined 28 Sertoli cell-specific marker products to clarify the different effects of single- and re-irradiation-induced infertility. It is well known that Sertoli cells are the somatic cells of the testes, supporting the architectural stability of germ cells and conserving the microenvironment and BTB [27,28,29,30]. The testis is a radiosensitive organ, and significant impairment of testis function could be caused by very low dose radiation [31]. In particular, with the breakdown of BTB by irradiation, germ cell autoantigens inside BTB may leak out repeatedly, leading to a continuous supply of autoantigens for immune stimulation with resultant anti-sperm antibody production and the prolongation of testicular inflammation [6]. In this study, 13 mRNA species (*Amh,*
*Claudin-11,*
*Dhh, Espn, Fshr, Fyn, GATA1, Inhbb**, Msi1, Spata2, Sox9, Trf,* and *Wt1*) showed significantly decreased expression (Figure 3A) in response to single- and re-irradiation, accompanied by significantly higher serum IgM levels in mice after single- and re-irradiation. Furthermore, among the above decreased thirteen mRNA species, six mRNA species (*Claudin-11,*
*Espn, Fshr, GATA1, Inhbb**,* and *Wt1*) showed significant differences between single- and re-irradiation treatments. Compared to group I, only *C**laudin**-**11* expression was significantly decreased in 4 GyII and 6 GyII mice and significantly decreased expression of *Espn* and *Inhbb* was only found in 4 GyII mice. On the other hand, the expression of *Fshr, GATA1,* and *Wt1* was significantly increased in 6 GyII mice compared to 6 GyI mice. Furthermore, four Sertoli cell-specific mRNA species (*Tjp1, occludin, ZO*-1, and *ZO*-2) were detected only in re-irradiated mice but not in single-irradiated mice (Figure 3C,D). Various integral tight junction proteins have been described in Sertoli cell junctions, including members of the claudin family (claudin-1 [32], claudin-3 [33,34,35], claudin-5 [36], claudin-12 [37], claudin-11 [38,39], claudin-13 [35], and occludin [40,41]). These proteins are linked to the actin cytoskeleton via cytoplasmic plaque proteins, including zonula occludens-1, -2, and -3 (ZO-1, ZO-2, and ZO-3) and provide links to the gap- and adherens-junctional types in the BTB. In these junction proteins, only claudin-11 is essential for spermatogenesis, as the claudin-11 knockout is infertile [38,39]. In this study, we demonstrated that only *C**laudin**-**11* expression was significantly decreased in all re-irradiated mice compared to that in single-irradiation mice. This indicates that it may be an important factor associated with testicular aspermatogenesis induced by irradiation. Concurrently, the transformation variability of these Sertoli cell-specific markers after single- or re-irradiation may be the reason for the significantly higher serum IgM levels in 6 GyI mice.

In particular, two decreased Sertoli cell-specific mRNA species *(**Aqp8* and *Clu**)* after single irradiation were significantly increased in re-irradiated mice (Figure 3C). Furthermore, in all five increased expressions of Sertoli cell-specific mRNA species, only *Testin* showed a significant increase in re-irradiated mice compared to that in single-irradiation mice (Figure 3B). It has been demonstrated that Testin localizes on the Sertoli cell surface that contacts germ cells, and the depletion of germ cells leads to an increase in *Testin* mRNA level [42]. Some studies have demonstrated that the *AQP8* transcript is expressed in Sertoli cells and occurs in all stages of spermatogenesis [43,44,45]. Similar reports have demonstrated that clusterin (*Clu*) is secreted by Sertoli cells and deposited onto the membranes of elongating spermatids and mature spermatozoa [46,47]. Recently, we demonstrated that *Testin* and *Clu* mRNA species increased with germ cell depletion by busulfan treatment and restored with completely recovered spermatogenesis [48]. In the present study, we further demonstrated that *AQP8,*
*Testin*, and *Clu* mRNA species were significantly increased following germ cell depletion by re-irradiation treatment. Some studies have suggested that Sertoli cells play a key role in mediating testicular immunology in irradiation-induced aspermatogenesis, while our study showed the altered Sertoli cell function after irra-treatment with intact spermatogenesis. These findings suggest that these Sertoli cell factors are more intimately interrelated with spermatogenesis after radiotherapy, providing a basis for the estimation of male fertility. 

This is the first report that compared the germ cell- and Sertoli cell-specific mRNA species after single- or re-irradiation treatment in mice. Although the aspermatogenesis after radiotherapy have been extensively studied, there is little information regarding the noxious role of single- and re-irradiation-induced male infertility. Knowledge of this impaired testicular immunopathologic microenvironment will be useful to understand infertility as adverse effects of radiotherapy. The above limited experimental data would address further information about this important clinical problem.

## Figures and Tables

**Figure 1 genes-13-00151-f001:**
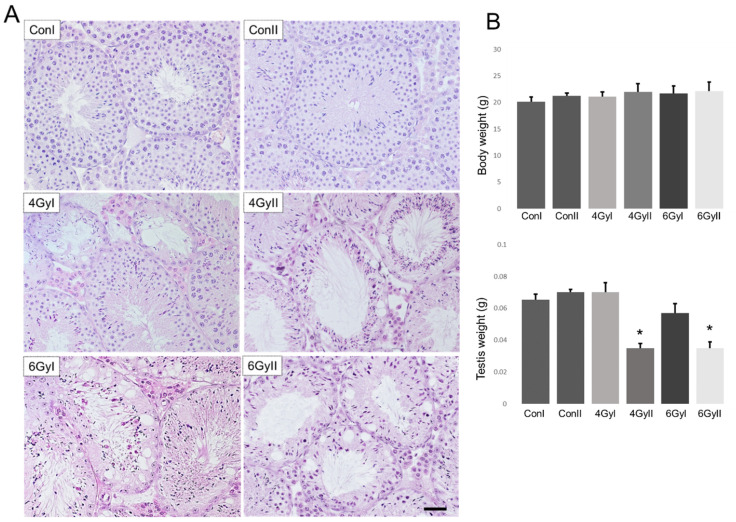
Testicular histology, testis weight, and body weight in each group (*n* = 5). (**A**) Testes sections show morphology in groups I and II. Intact seminiferous tubules showing normal germinal epithelium from the spermatogonia to spermatozoa are observed in ConI and ConII. Damaged seminiferous tubules with azoospermia are observed in the other groups of 4GyI, 4GyII, 6GyI, and 6GyII. The presence of both atrophic and intact seminiferous tubules with spermatogenesis was observed in the group IV mice. Normal-appearing seminiferous tubules are observed in the group V mice. Scale bar = 50 µm. (**B**) The body weight and testis weight in each group. * *p* < 0.05 vs. control group.

**Figure 2 genes-13-00151-f002:**
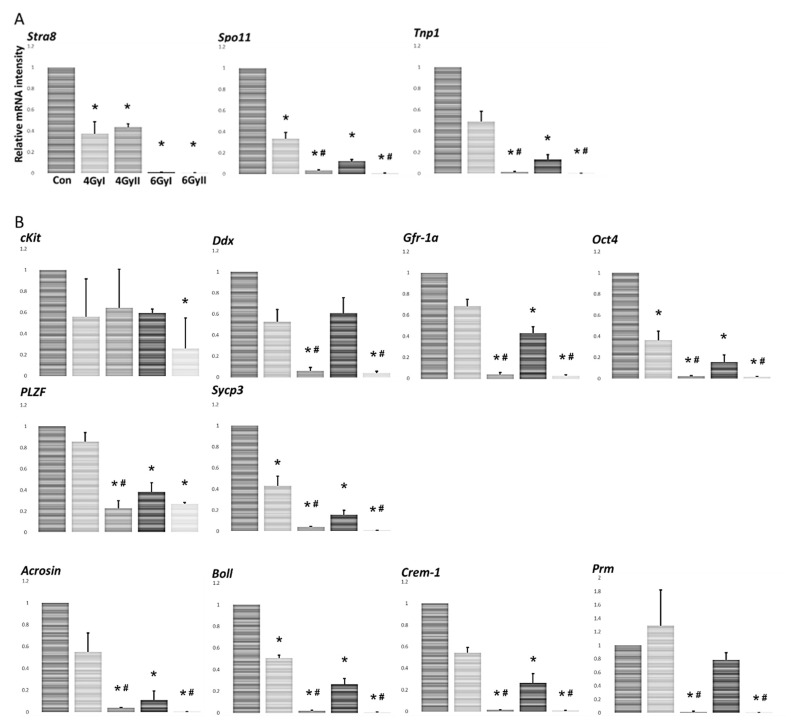
The levels of mRNA transcripts encoding markers of germ cell differentiation in each group (*n* = 5). Expression was measured by real-time RT-PCR, and the results were expressed relative to the internal control GAPDH. (**A**) Data show expression of the spermatogonial marker Stra8, the spermatocyte marker Spo11, and the spermatid marker Tnp1 expressed as the mean ± SD of five mice in each group. The y-axis shows the relative mRNA intensity. (**B**) The expression of the premeiotic markers (cKit, Gfra1, Oct4, PLZF, Sycp3, and Ddx4) and the meiotic and postmeiotic markers (Boll, Crem, Prm1, and Acrosin). The results are expressed as the mean ± SD of five mice in each group, and the y-axis shows the relative mRNA intensity. * *p* < 0.05 vs. control group; # *p* < 0.05, vs. group I.

**Figure 3 genes-13-00151-f003:**
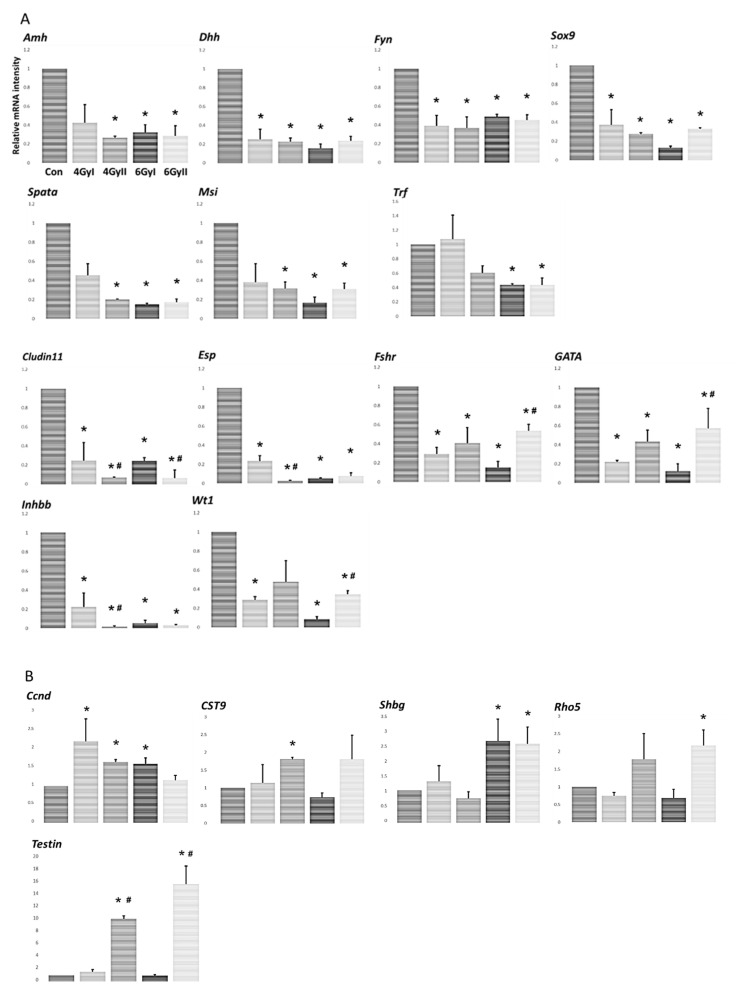
The levels of 27 mRNA transcripts encoding markers of Sertoli cell-specific products in each group. Expression was measured by real-time RT PCR, and the results are expressed relative to the internal control GAPDH, expressed as the mean ± SD of five mice in each group. The y-axis shows relative mRNA intensity. (**A**) Transcripts showing a significant decrease in levels after irradiation treatment compared to control values. (**B**) Transcripts showing a significant increase in levels after irradiation treatment compared to control values. (**C**) Transcripts showing a significant decrease in levels only in group I but not group II compared to control values. (**D**) Transcripts showing a significant decrease in levels only in group II but not group I compared to control values. (**E**) Transcripts showing no change in levels after irradiation treatment compared to control values. * *p* < 0.05 vs. control group; # *p* < 0.05 vs. group I.

**Figure 4 genes-13-00151-f004:**
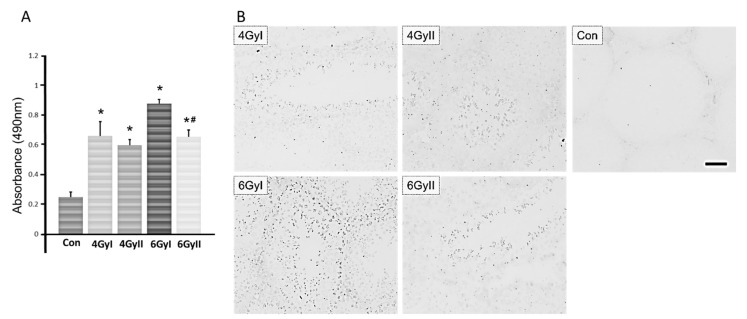
Detection of serum anti-germ cell antibody levels in each group. (**A**) ELISA measurements of autoantibodies in each group reactive with germ cells antigens. The results are expressed as the mean ± SD of five mice in each group. * *p* < 0.05 vs. control group; # *p* < 0.05 vs. group I. (**B**) Normal frozen sections of seminiferous tubules were reacted with diluted sera obtained from each group mice followed by incubation with horseradish peroxidase (HRP)-conjugated anti-mouse IgM. Scale bar, 50 mm.

**Table 1 genes-13-00151-t001:** Primers used for real-time RT-PCR.

Gene	Forward Primer	Reverse Primer
*GAPDH*	TGTGTCCGTCGTGGATCTGA	TTGCTGTTGAAGTCGCAGGAG
Sertoli cell		
*Amh*	TCCTACATCTGGCTGAAGTGATATGGG	AGGTTCTGTGTGCCCCGCAG
*Aqp8*	GCTGGCAGTCACAGTGATCGGA	CCTGGACGATGGCAAAGGCTG
*Ccnd2*	GGAACCTGGCCGCAGTCACC	AATCATCGACGGCGGGTACATG
*Claudin-11*	TCACAACGTCCACCAATGACTG	GGCACATACAGGAAACCAGATG
*Clu*	CCACGCCATGAAGATTCTCCTGC	CTCCCTGGACGGCGTTCTGA
*Cst12*	GGATGACGATTTTGCCTACAAGTTCCT	TTCTCTCTCCTGGACCTTCCTGCA
*Cst9*	GATATTTGCCCCTTTCAGGAGAGCC	AGAGAAGTACGTGACCAGTCCATGGG
*Dhh*	GGCGCAGACCGCCTGATG	AAGGCACGGCCTTCGTAGTGG
*Espn*	GCTTCTGGTCGGGCATTACCCT	GTGTCATGCCGTCTTGGGCG
*Fshr*	GGCCAGGTCAACATACCGCTTG	TGCCTTGAAATAGACTTGTTGCAAATTG
*Fyn*	GAAGCGGCCCTGTATGGAAGGTT	TGTGGGCAGGGCATCCTATAGC
*GATA1*	ATGGTCAGAACCGGCCTCTCATC	GAGCTTGAAATAGAGGCCGCAGG
*Inhba*	CATGGAGCAGACCTCGGAGATCA	TGGTCCTGGTTCTGTTAGCCTTGG
*Inhbb*	GAGCGCGTCTCCGAGATCATCA	CGTACCTTCCTCCTGCTGCCCTT
*Msi1*	TCACTTTCATGGACCAGGCGG	GTTCACAGACAGCCCCCCCA
*Occlidin*	CTTCTGCTTCATCGCTTCC	CTTGCCCTTTCCTGCTTTC
*Rhox5*	AGGTTCGCCCAGCATCGACTG	GCCGCAGCCCTCCTGATCTT
*Shbg*	GACATTCCCCAGCCTCATGCA	TGCCTCGGAAGACAGAACCACG
*Sox9*	CGCGGAGCTCAGCAAGACTCTG	TGTCCGTTCTTCACCGACTTCCTC
*Spata2*	GCCGTGTGGGCCTGTGCTT	TTCCCCAAATCAAACCCAAGGG
*sympk*	CAAGAAGAAGGGCCAAGCATCGA	AGGAAGTTGTCAAGCAGGGTGGG
*Testin*	AAAGACAATGGCGGCCTCGC	GGCCCCACTTTAGCCACTGCC
*Tjp1*	GCGGAGAGAGACAAGATGTCCGC	CTCTGAAAATGAGGAT- TATCTCTTCCACCA
*Trf*	CAAATGCATCAGCTTCCGTGACC	CGGCATCGTACACCCAACCC
*Wnt5a*	CTGCTTCTACCATGCGTTTGCTGG	GCCATGGGACAGTGCGGC
*Wt1*	GCTCCAGCTCAGTGAAATGGACAGAA	GGCCACTCCAGATACACGCCG
*ZO-1*	ACAAACAGCCCTACCAACC	CCATCCTCATCTTCATCTTCTTC
*ZO-2*	GTTTTTCTTCGTCCTAGTCCC	CATCCATCCCTTCCATCTTTC
Germ Cell		
*Acrosin*	TGTCCGTGGTTGCCAAGGATAACA	AATCCGGGTACCTGCTTGTGAGTT
*Boule*	AACCCAACAAGTGGCCCAAGATAC	CTTTGGACACTCCAGCTCTGTCAT
*c-Kit*	GCATCACCATCAAAAACGTG	GATAGTCAGCGTCTCCTGGC
*CREM-1*	TTCTTTCACGAAGACCCTCA	TGTTAGGTGGTGTCCCTTCT
*Ddx4*	AAGCAGAGGGTTTTCCAAGC	GCCTGATGCTTCTGAATCG
*Gfr-α1*	CAGTTTTCGTCTGCTGAGGTTG	TCTGCTCAAAGTGGCTCCAT
*OCT4*	TGCGGAGGGATGGCATACTG	GCACAGGGCTCAGAGGAGGTTC
*Protamine*	TCCATCAAAACTCCTGCGTGA	AGGTGGCATTGTTCCTTAGCA
*Spo11*	CGCGTGGCCTCTAGTTCTGAGG	GGTATCATCCGAAGGCCGACAGAAT
*Stra8*	GAAGGTGCATGGTTCACCGTGG	GCTCGATCGGCGGGCCTGTG
*Sycp3*	TGGAGCTGACATCAACAAAGC	CCCACTGCTGCAACACATTC
*Tnp1*	GGCGATGATGCAAGTCGCAA	CCACTCTGATAGGATCTTTGGCTTTTGG
*Zbtb16*	AACGGTTCCTGGACAGTTTG	CCACACAGCAGACAGAAGA

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
