# Peer review of "Changes in Expression of Specific mRNA Transcripts after Single- or Re-Irradiation in Mouse Testes"

_genes, 2022, doi:10.3390/genes13010151_

Round 1
Reviewer 1 Report
Nagahori et al. describe observed changes in expressions of several testes-specific genes in the context of no, single, or double irradiation with the intention of investigating whether double irradiation significantly affects the gene regulation and development/maturity of the testicular cells. This is an important clinical problem to be addressed. The results were clearly explained. I only have comments related to the manuscript writing and presentation itself:
- On line 42, when the authors say "rescue", do they mean rescue of aspermatogenesis that is induced by irradation compared to busulfan-induced? It sounds like they mean to talk about irradiation here. If so, they should mention "irradiation" to make that clear.
- In all results sections (3.1 to 3.4), the authors do a good job of explaining each figure result. However, they should start each section with an explanation for what they were trying to accomplish and why so the reader understand the rationale for why the investigation was done. At the end of each section, the authors should also summarize the overall main findings of the section so that the reader understands why those results are important.
- In figures 2 - 4, the authors compare both irradiated groups I and II with "con; control group". Why did they not compare 4/6GyI with ConI and 4/6GyII with ConII as they did in Figure 1?
- (Minor comment) On line 156, it appears that the author mistakenly added the word "also" at the beginning of the sentence. That word does not appear to be needed.
Author Response
Responses to Reviewer 1’s comments:
Nagahori et al. describe observed changes in expressions of several testes-specific genes in the context of no, single, or double irradiation with the intention of investigating whether double irradiation significantly affects the gene regulation and development/maturity of the testicular cells. This is an important clinical problem to be addressed. The results were clearly explained. I only have comments related to the manuscript writing and presentation itself.
Answer: We would like to thank you for your evaluation. Many thanks for your very valuable and constructive feedback points.
Comments 1: On line 42, when the authors say "rescue", do they mean rescue of aspermatogenesis that is induced by irradation compared to busulfan-induced? It sounds like they mean to talk about irradiation here. If so, they should mention "irradiation" to make that clear.
Answer: Thanks for your comments, we have added the sentences “in irradiation-mice” in the entire manuscript (see lines 42).
Comments 2: In all results sections (3.1 to 3.4), the authors do a good job of explaining each figure result. However, they should start each section with an explanation for what they were trying to accomplish and why so the reader understand the rationale for why the investigation was done. At the end of each section, the authors should also summarize the overall main findings of the section so that the reader understands why those results are important.
Answer: Thank you for your evaluation. We agree with the comments and have added the explanation at the start of each section and the summary at the end of each section in all sections 3.1 to 3.4 (see lines 153-155; 162-164; 173; 183-185; 202-203; 214-218; 229-230).
Comments 3: In figures 2 - 4, the authors compare both irradiated groups I and II with "con; control group". Why did they not compare 4/6GyI with ConI and 4/6GyII with ConII as they did in Figure 1?
Answer: Thanks for your comments. In Figure 1, we demonstrated that the normal non-irradiated mice showed spermatogenesis maturity and increased body and testis weights from the ages of 5 weeks (ConI) to 6 weeks (ConII), the control group was separated as ConI and ConII. In Figure 2-3, we analyzed mRNA species in testes using real-time RT-PCR and calculated the relative mRNA intensity to the amount of GAPDH transcript used as an internal control. The mRNA species expression in the control group (ConI and ConII) for each point was normalized to 1. Furthermore, there are no difference in ELISA analysis using sera (1:160 dilution) from normal mice of 5 weeks (ConI) or 6 weeks (ConII), so only Con data was showed in Figure 4. We appreciate your understanding.
Comments 4: (Minor comment) On line 156, it appears that the author mistakenly added the word "also" at the beginning of the sentence. That word does not appear to be needed.
Answer: We agree with the comments and corrected the sentence (see lines 157-158). Thank you very much.
Reviewer 2 Report
The paper entitled “Changes in the expression of specific mRNA trascripts after single or re-irradiation in mouse testes” is a substantial analysis about the effects of re-irradiation in mouse testes. The study represents an important starting point to continue investigating the noxious role of re-irradiation in testes. There are only some few questions about this work.
- The irradiation doses was explained in the Introduction section (Lines 53-56) but it would be useful to describe this crucial point in the Materials and Methods section.
- The experiments conducted by the authors have important clinical implications. The discussion is well conducted considering the experimental data, but what about the clinical information present in literature? Please, discuss this aspect citing the clinical studies published on this matter.
- It would be helpful to discuss the strengths and the weaknesses of this study in the “Discussion” section.
Author Response
Responses to Reviewer 2’s comments:
The paper entitled “Changes in the expression of specific mRNA trascripts after single or re-irradiation in mouse testes” is a substantial analysis about the effects of re-irradiation in mouse testes. The study represents an important starting point to continue investigating the noxious role of re-irradiation in testes. There are only some few questions about this work.
Answer: We would like to thank you for your evaluation. Many thanks for your very valuable and constructive feedback points.
Comments 1: The irradiation doses was explained in the Introduction section (Lines 53-56) but it would be useful to describe this crucial point in the Materials and Methods section.
Answer: Thanks for your comments, we have added the sentences “In the present study, we examined the impaired testicular functions induced by low-dose radiation exposure in mice to address the immune-pathophysiological differences between single- or re-irradiation.” in the entire manuscript (see lines 75-77).
Comments 2: The experiments conducted by the authors have important clinical implications. The discussion is well conducted considering the experimental data, but what about the clinical information present in literature? Please, discuss this aspect citing the clinical studies published on this matter.
Comments 3: It would be helpful to discuss the strengths and the weaknesses of this study in the “Discussion” section.
Answer: Thank you for your evaluation. We agree with the comments and have added” This is the first report compared the germ cell- and Sertoli cell-specific mRNA species after single- or re-irradiation treatment in mice. Although the aspermatogenesis after radiotherapy have been extensively studied, there is little information regarding the noxious role of single- and re-irradiation induced male infertility. Knowledge of this impaired testicular immunopathologic microenvironment will be useful to understand infertility as adverse effects of radiotherapy. The above limited experimental data would address the further information about this important clinical problem.” in the entire manuscript (see lines 339-345).